# Path Analysis to Assess Socio-Economic and Mitigation Measure Determinants for Daily Coronavirus Infections

**DOI:** 10.3390/ijerph181910071

**Published:** 2021-09-25

**Authors:** Elie Yammine, Abbas Rammal

**Affiliations:** 1Statistics and Computer Science Department, Faculty of Science, Lebanese University, Baabda 1003, Lebanon; e.yammine@st.ul.edu.lb; 2Data Science Department, Faculty of Information, Lebanese University, Baabda 1003, Lebanon

**Keywords:** path analysis, COVID-19 pandemic, direct and indirect effects, socio-economic factor, mitigation factor

## Abstract

(1) Background: With the rapid global spread of the coronavirus disease 2019 (COVID-19) and the relatively high daily cases recorded in a short time compared to other types of seasonal flu, the world remains under continuous threat unless we identify the key factors that contribute to these unexpected records. This identification is important for developing effective criteria and plans to reduce the spread of the COVID-19 pandemic and can guide national authorities to tighten or reduce mitigation measures, in addition to spreading awareness of the important factors that contribute to the propagation of the disease. (2) Methods: The data represents the daily infections (210 days) in four different countries (China, Italy, Iran, and Lebanon) taken approximately in the same duration, between January and March 2020. Path analysis was implemented on the data to detect the significant factors that affect the daily COVID-19 infections. (3) Results: The path coefficients show that quarantine commitment (β = −0.823) and full lockdown measures (β = −0.775) have the largest direct effect on COVID-19 daily infections. The results also show that more experience (β = −0.35), density in society (β = −0.288), medical resources (β = 0.136), and economic resources (β = 0.142) have indirect effects on daily COVID-19 infections. (4) Conclusions: The COVID-19 daily infections directly decrease with complete lockdown measures, quarantine commitment, wearing masks, and social distancing. COVID-19 daily cases are indirectly associated with population density, special events, previous experience, technology used, economic resources, and medical resources.

## 1. Introduction

The coronavirus disease 2019 (COVID-19) has rapidly spread around the world since it first appeared in the city of Wuhan, China, towards the end of December 2019. The World Health Organization classified it as a pandemic in March 2020 [1]. Europe and the U.S. have recorded the highest number of infections and deaths. U.S. cases formed more than one fourth of total global infections by June 2020 [2]. Governmental and institutional reactions and measures varied across countries with respect to the time of introduction of social distancing measures and their degree of severity. Globally, these control measures have caused significant disruption to social and economic structures. However, it is unknown whether these policies have had an impact, and how long they should remain in place. It is thus essential to assess the effects of these control measures on the pandemic for the benefit of global health security.

Although there have been many efforts to analyze and predict the behavior of CVOID-19 infections, due to the highly complex nature of the outbreak and the variation in its behavior from nation-to-nation, the main challenge is to determine the factors that affect the increase of daily infections. The study is aimed at examining the socio-economic and mitigation measure determinants of the daily infections. Through the path analysis technique, which is a form of multiple regression statistical analysis that is used to evaluate causal models by examining the relationships between a dependent variable and independent variables, we can estimate both the magnitude and significance of causal connections between variables [3]. For this reason, we apply the path analysis technique in this study, which involves the analysis of hypothesized relationships among multiple variables [4]. This technique consists of a family of models that depicts the influence of a set of variables on one another [5].

Many studies have focused on the application of path analysis on the COVID-19 pandemic. V. Burkova conducted a path analysis to examine possible factors that may be associated with self-reported levels of anxiety during the first wave of the COVID-19 pandemic [6], while B. Wielgus performed a path analysis to examine the relationship between anxiety and general psychosomatic functioning during the COVID-19 pandemic, considering the influence of indirect factors such as psychological flexibility and mindfulness [7]. Annette Brose performed multilevel structural equation modeling to identify mechanisms underlying changes in well-being in times of threat in the COVID-19 pandemic, with a focus on appraisals of the pandemic and affective states, stress, and mindfulness in daily life [8]. On the other hand, L. Tamariz conducted structural equation modeling (SEM) on COVID-19 infections in South Florida and found that the infection is associated with economic disadvantage in a particular geographical area and not with racial/ethnic distribution [9]. Furthermore, M. Zareipour conducted a study based on a path analysis to find the determinants of COVID-19 prevention behavior in the elderly in Urmia, Iran, and found that effective interventions based on the health belief model and promoting knowledge, perceived susceptibility, severity, and perceived self-efficacy can prevent the elderly from contracting this disease [10]. Marvin G. Pizon generated a path analysis model of COVID-19 to establish the specific cause-and-effect between air pressure, air temperature, and relative humidity [11]. L. Salehi applied a path analysis to assess the relationship of fear and anxiety caused by COVID-19 with pregnancy and the mental health of pregnant women and found that it is necessary to pay more attention to the mental health of pregnant women during the pandemic [12].

Path analysis has also been used widely in the medical field. Hardenberg developed a path analysis model based on linear equation system for use in phylogenetic studies [13]. In his article, using a path analysis, H. Nadrian examined the possible direct/indirect effects of health belief model (HBM) constructs on self-care behaviors among heart failure patients [14]. Rebekah J. Walker studied the association between the social determinants of health to outcomes in individuals with type 2 diabetes and the results were consistent with a previous conceptual framework which stated that there exist a direct and an indirect link between socio-economic and psychosocial factors and glycemic control [15]. Path analysis and SEM are some of the most used techniques nowadays despite the continuous rise of new and sophisticated methods in social and medical sciences.

Previous studies have shown that many factors can be associated to the daily cases of COVID-19. It was found in Thailand that touristic and cultural activities are significant factors that contribute to the number of COVID-19 cases [16]. In Italy, a strict lockdown decreased the transmission rate to maintain societal immunity [17]. Population density is found to be positively related with deaths due to COVID-19 in low populated countries [18]. Income, social capital, and trust and beliefs are proven to be significant factors related to daily COVID-19 cases [19].

The article is organized as follows. In Section 2, we explain the variables used in the study, how data were collected, and the statistical methodology used. Second, we present the results and coefficients in Section 3. Finally, we analyze and interpret our results and provide the discussion and conclusions in Section 4 and Section 5, respectively. 

## 2. Materials and Methods

### 2.1. Conceptual Framework

#### 2.1.1. Statement of Problem

As the world witnessed a continuous increase in the daily infections of COVID-19 with great fear of an uncontrolled spread of the disease, it became essential to determine the variables and factors affecting this increase and take immediate action to control the spread.

#### 2.1.2. Importance of Variables Selected

Because COVID-19 transfers through surfaces and the air, implementing lockdown measures with different levels is important to reduce contact between people. Quarantine and lockdowns have always been effective ways to control communicable disease outbreaks. An example of this is the 2003 SARS outbreak, where the use of quarantine, border controls, contact tracing, and surveillance proved to be effective in containing the global threat in just over three months [20]. We also think that medical and economic resources are among the main factors that contribute to the daily COVID-19 cases where countries with enough resources will not face difficulties in controlling the spread, whereas a lack of resources has been a source of weakness in fighting against such SARS diseases. Financing profoundly affects the performance of the health system in a specific country. Any policy that the health system decides to implement or not directly depends on the amount of available funding [21]. Experience in dealing with health outbreaks greatly impacted how countries in response to COVID-19, as in the case of Hong Kong, which faced the 1957 “Asian” and 1968 “Hong Kong” influenza pandemics, along with A(H7N9) in 2013. In addition, Taiwan experienced the SARS outbreak in 2003, whereas Liberia was profoundly affected by the Ebola epidemic in 2014, which led to thousands of deaths. 

All these experiences made local governments realize the importance of establishing a tiered command structure to prepare for and respond to future outbreaks and consolidate all health protection functions. As a result, the public health systems and social measures in Hong Kong proved to be critical in controlling COVID-19. Liberia maintained a low level of spread of the COVID-19 while Taiwan recorded only around 600 positive cases by March 2021 [22]. Some studies showed that the population density is important in modeling the COVID-19 infections. A study in the U.S. revealed that population density is an effective predictor of cumulative infection cases at the country level [23]. Other studies [24,25,26,27] have shown that SARS-CoV-2 transmission is potentially more likely to occur among cities with higher population densities. The use of modern technology in healthcare systems has helped in many aspects. Artificial Intelligence (AI) is used to identify, track, and forecast outbreaks and help in diagnosing the virus. It is used in processing the healthcare claims. Drones and robots are used to deliver food and medical supplies and sterilize public places. AI is helping to develop drugs and COVID-19 vaccines through the use of supercomputers [28].

Not all factors influence the COVID-19 daily cases directly. For that we assume some variables have direct and indirect effects, either negative or positive. Performing a series of multiple regressions among the independent variables can help us identify the mediators that connect independent variables with the daily COVID-19 cases. Based on the multiple regressions, mediators are included in the path analysis so that independent variables have direct and indirect effects on our dependent variable.

### 2.2. Data Set Collection

To answer and judge the test hypothesis and evaluate the outcomes of particular questions, we used the process of collecting and measuring data. Thus, to predict the behavior of the spread of coronavirus, four countries were chosen that adopted different methodologies to deal with the COVID-19 pandemic and achieved different results related to the methodology used: China (67 days), Lebanon (40 days), Italy (61 days), and Iran (42 days).

After studying the situation of the virus in these countries, we noticed several indicators that directly or indirectly affected the level of spread in each country. These are:The governments’ reactions: this factor refers to the different responses and reactions from the governments of the four countries during the outbreak. These indices are used to explore whether government response affects the rate of infection and identify correlates of intense responses.The medical resources: this factor refers to the health system policies such as the COVID-19 testing regime or emergency investments into healthcare (ICU beds, etc.) and the health services quality in these four countries. The sensitivity effects of this factor on the results are proposed to be investigated in this study.The commitment of the people in each country to government guidelines. Theoretically, this factor must have a direct dependance on the intensity of the spread of COVID-19 in these countries.The special events: this factor takes into consideration the existence of simultaneous events that affected the spread of COVID-19: other disasters, economic problems, war, political problems or disturbance, official holidays, etc.The economic level and governmental aids: this factor refer to the economic policies enacted during the pandemic, such as income support to citizens or the provision of foreign aid. Depending on the direct relationship between this factor and the quarantine compliance of the people in each country, we have proposed it to be present in this study.Previous experience in the history of the four governments that determined the existence of experience in critical disaster management, or lack thereof.The use of technology devoted to control the virus spread in these countries to help in health and hospitalization services, lockdown control, and restrictions of infected zones.The population density: this is considered as the number of the people per 1 km^2^ in the four countries, which can affect the spread of the virus.The family number: this refers to the average number of family members in each of the selected countries.

These direct and indirect factors were used as parameters by our model to predict the future behavior of the spread of COVID-19. The data were combined into a series of novel indices that aggregated various measures of each factor. The parameters were then measured and detected depending on specified criteria and are presented in Table 1.

Thus, we collected real data (210 days) of the four countries, from different official sources for precise parameters and the daily infection records. The date range for the data for each of the four countries is shown below.

China: 9 January 2020–28 March 2020 (80 days)Lebanon: 21 February 2020–31 March 2020 (40 days)Italy: 31 January 2020–31 March 2020 (61 days)Iran: 19 February 2020–31 March 2020 (42 days)

The dependent variable is the daily infections records which are basically the cumulative records for a dependent day-by-day scale. In other words, the record of the next day is the sum of the records of the current day and the new records obtained in the same day. Table 1 shows the technique for coding each of the factors by developing the measurement scales used to build the model. This is the first basic step to build and develop a model using the structural equation modeling method.

In addition, the data collected do not include missing values. Using the free missing values factors, we can predict the degrees of the possibility of COVID-19 infection with the help of a machine learning algorithm. These methods may result in better accuracy, unless a missing value is expected to have a very high variance.

### 2.3. Hypothesis

We hypothesized that lockdown, medical resources, economic resources, technology used, population density, previous experience, family number, procedure, and special events variables influenced the daily COVID-19 infections. 

The purpose of studying the above hypothesis lies in determining the factors with the most influence on the development of the COVID-19 pandemic. This will enable us to act quickly and consciously in tightening or reducing the mitigation measures, thereby leading to a better understanding of the behavior of the virus. 

### 2.4. Statistical Analysis

The data were analyzed by path analysis using the AMOS and SPSS statistical software to determine the direct and indirect effects. We used structural equation modeling (SEM) which is defined as a combination of factor analysis and regression. SEM is a powerful, multivariate technique used increasingly in scientific investigations to test and evaluate multivariate causal relationships. SEM differs from other modeling approaches in that it tests the direct and indirect effects on pre-assumed causal relationships. Path analysis was developed to quantify the relationships among multiple variables [29]. It was the early name for SEM before there were latent variables, and it was very powerful in testing and developing the structural hypothesis with both indirect and direct causal effects. However, the two effects have recently been synonymized. Path analysis can explain the causal relationships among variables. A common function of path analysis is mediation, which assumes that a variable can influence an outcome directly or indirectly through another variable. The interest in SEM is generally on constructs called latent variables. The relationship between the latent variables is represented by regression or path coefficients. The structural equation model implies a structure of the covariances between the observed variable and the latent variable [30]. Path analysis is a statistical technique that uses both bivariate and multiple linear regression techniques to test the causal relations among the variables specialized in the model [31]. By using this method, we can estimate both the magnitude and significance of causal connections between variables. In this study, path coefficients were computed via a series of multiple regression analyses based on the hypothesized model. Path diagrams were constructed with a single headed arrow representing the causal order between two variables, with the head pointing to the effect and the tail to the cause. A curved, double arrow indicated a correlation between two variables. The method is also known as causal modeling, analysis of covariance structures, and latent variable model [32]. The sample size in this study was adequate based on the recommendation by Kline [6] that 10–20 times as many cases as parameters are sufficient for significance testing of model effects.

Path analysis is comprised of four stages: (1) model specification: statement of the theoretical model in terms of equations or a diagram; (2) model identification and parameter estimate: the theoretical model can be estimated with observed data. The model’s parameters are statistically estimated from data. Multiple regression is one such estimation method, but most often more complicated methods are used; (3) model fit: the estimated model parameters are used to predict the correlations or covariances between measured variables and the predicted correlations or covariances are compared to the observed correlations or covariances; (4) model respecification: the model is respecified by adding or removing a significant or a non-significant parameter estimate depending on its P-value and the change of the chi-square of the model. The final process of the path analysis is the resulting identification of the effects of independent variables on the dependent variable. The relationship between the variables is described in the form of structural equations. The structural equations are constructed by calculating the direct effects (DE), indirect effects (IE), and the total effect (TE) between the variables [33]. The values of these indices are determined based on the path coefficients. The stages of path analysis are depicted in Figure 1.

## 3. Results

### 3.1. Descriptive Analysis

The days were distributed as follows: 31.9% from China, 20% from Iran, 29% from Italy, and 19% from Lebanon. Only China had previous experience in dealing with a viral outbreak, whereas the remaining countries had no experience. Most days (39%), people were not completely committed to quarantine measures, and there were no special events 49% of the days. A full lockdown was held 43.8% of the days. Moreover, on 47% of the days, medical resources were considered as good, and on 50.5% of the days, economic resources were low. The technology used was considered low and high 30% of the time, respectively.

Table 2 displays a summary statistic on the variables used. We can see from Table 2 that in most days medical resources (mean = 0.7774) were available, whereas there was not enough technology available to use for mitigation measures (mean = 0.4679). 

Figure 2 shows that China, which implemented a full lockdown, recorded as much COVID-19 cases as other countries that implemented partial lockdowns. Meanwhile, Iran recorded less cases than Italy with the same lockdown measures (Lockdown = 0.25, 0.5).

Figure 3 shows that countries with prior experience in health crises were able to reduce the transmission of COVID-19, whereas countries with no previous experience recorded higher cases of the virus.

Table 3 displays the Pearson correlation between the independent variables among each other and between every independent variable and the daily cases of COVID-19.

Table 3 shows that there exist five variables (Technology Used, Procedure, Density, Medical Resources, and Economic Resources) that are correlated with the daily cases at a 0.01 level of significance, while only one variable (Family number) is correlated to the daily cases with 0.05 level of significance. Because 6 out of 10 variables are significantly correlated to the daily cases of COVID-19, and since we are trying to assess and detect the factors that most contribute to the daily infections, path analysis is the right methodology to use. Most variables are correlated to each other which creates dependencies and associations among the independent variables; thus, mediators (variables carrying the indirect effects) have high correlations with the independent ones.

### 3.2. Evaluation of Path Analysis

First, we present the necessary indices that validate our path model. One of the most used fit indices worldwide is the Chi-square goodness of fit resulting from maximum likelihood estimation (MLE). In fact, the smaller $\chi_{GOF}^{2}$ is, the better the fit model. In our model, a minimum Chi-square of 83.1 was reached after 11 iterations. The probability level obtained was equal to 0, verifying the significance of the model. We consider two other goodness-of-fit indices: Akaike’s information criterion (AIC) and Schwarz Bayesian information criterion (BIC). These indices are not used to test the model in the sense of hypothesis testing, but for model selection. Given a data set, a researcher chooses either the AIC or BIC, and computes it for all models under consideration. Then, the model with the lowest index is selected. Note that both the AIC and BIC combine absolute fit with model parsimony [34]. The lowest AIC and BIC found are 157.1 and 280.943, respectively. The corrected Akaike’s information criterion (CAIC) = 317.943 The goodness-of-fit index (GFI), the proportion of variance accounted for by the estimated population covariance, is equal to 0.934. It is categorized as an absolute fit index (AFI) which examines the level of correspondence between the proposed model and the observed data. 

The following indices, called incremental fit indices, permitted us to evaluate the contribution of the estimated model with respect to the reference model (null model). These indices suggested improvements in the fit of the model. The comparative fit index (CFI), comparing the fit of a target model to the fit of an independent or null model, was equal to 0.978 for our model. The Tucker–Lewis index (TLI), used to measure a relative reduction in misfit per degree of freedom [35], was equal to 0.944. The normed fit index (NFI) which reflects the proportion by which a researcher’s model improves fit compared to the null model (uncorrelated measured variables) [36] was equal to 0.97. The relative fit index (RFI) is equal to 0.93 and the incremental fit index (IFI) is equal to 0.978. 

The better the model the more the above indices are close to 1. In our study, all the incremental fit indices were greater than 0.9 (cut-off value) [37], which verified that the model exists and is significant. The root mean square error of approximation (RMSEA), which is a supplementary statistic used to determine the fit to the Rasch model with a large sample size, was equal to 0.132. This was due to the small sample size of only 210 days. 

In our study, the number of measured variables (*k*) = 10, number of distinct sample moments $=\frac{\left(k\times \left(k+1\right)\right)}{2}=55$, and number of distinct parameters to be estimated = 37. The degree of freedom (df) = number of distinct sample moments − number of estimated parameters = 55 − 37 = 18 > 0 (overestimated). Thus, our hypothesis of whether the socio-economic and mitigation measure factors influenced daily COVID-19 infections could be tested via path analysis. 

The dependent variable was the daily COVID-19 infections. The exogenous variables were family number, procedure, special events, density, and previous experience. The endogenous variables were lockdown, medical resources, economic resources, and technology used. Error terms were considered as unobserved exogenous variables connected to the endogenous variables. Multicollinearity problems were absent since all bivariate correlations presented in Table 4 were below 0.8 [33]. Path coefficients (parameter estimates) were calculated based on the hypothesized model and the results are presented in Table 6.

As we expected, all factors had direct and indirect impact on the daily COVID-19 infections of varying strengths. However, there was an absence of significant causal effect from medical resources and economic resources to daily infections.

Table 5 shows that it is estimated that the predictors of medical resources explain 97.2% of its variance. In other words, the error variance of medical resources is approximately 2.8% of the variance of medical resources itself. Also, it is estimated that the predictors of lockdown explain 90.3% of its variance. In other words, the error variance of lockdown is approximately 9.7% of the variance of lockdown itself. The same interpretation applies for economic resources and technology used variables.

Table 6 shows that approximately all causal effects are significant with 95% confidence level. Although the causal relation between medical resources and lockdown is slightly not statistically significant, we still consider this relation in our model. 

Table 7 shows that due to the direct (unmediated) effect of procedure on daily Covid-19 cases, when procedure goes up by 1 standard deviation, daily covid-19 cases go down by 0.823 standard deviations (95% CI = −1.175 to −0.541; *p* < 0.05). Due to the direct (unmediated) effect of lockdown on daily covid-19 cases, when lockdown goes up by 1 standard deviation, daily covid-19 cases go down by 0.775 standard deviations (95% CI = −1.051 to −0.497; *p* < 0.05). Due to the direct (unmediated) effect of technology used on daily covid-19 cases, when technology used goes up by 1 standard deviation, daily covid-19 cases goes up by 0.17 standard deviations (95% CI = 0.015 to 0.287; *p* < 0.05).

The indirect effects of medical resources, special events, and lockdown with *p*-values respectively equal to 0.11, 0.12, and 0.976 are not statistically significant. Moreover, the confidence intervals (CIs) for the non-significant indirect effects contain zeros, which is strong evidence of the non-significance of these effects. Meanwhile, all other CIs do not contain zeros, which is strong evidence of their estimates’ significance. Due to the indirect (mediated) effect of experience on daily covid-19 cases, when experience goes up by 1 standard deviation, daily covid-19 cases go down by 0.288 standard deviations (95% CI = −0.475 to −0.122; *p* < 0.05). Due to the indirect (mediated) effect of procedure on daily covid-19 cases, when procedure goes up by 1 standard deviation, daily covid-19 Cases go up by 0.355 standard deviations (95% CI = 0.19 to 0.602; *p* < 0.05). Due to the indirect (mediated) effect of density on daily covid-19 cases, when density goes up by 1 standard deviation, daily covid-19 cases go down by 0.35 standard deviations (95% CI = −0.493 to −0.231; *p* < 0.05). Due to the indirect (mediated) effect of family number on Daily covid-19 cases, when family number goes up by 1 standard deviation, daily covid-19 cases go down by 0.097 standard deviations (95% CI = −0.155 to −0.023; *p* < 0.05). Due to the indirect (mediated) effect of economic resources on daily covid-19 cases, when economic resources go up by 1 standard deviation, daily covid-19 cases go up by 0.142 standard deviations (95% CI = 0.011 to 0.236; *p* < 0.05). Due to both direct (unmediated) and indirect (mediated) effects of procedure on daily covid-19 cases, when procedure goes up by 1 standard deviation, daily covid-19 cases go down by 0.468 standard deviations (95% CI = −0.649 to −0.314; *p* < 0.05). Due to both direct (unmediated) and indirect (mediated) effects of lockdown on daily covid-19 cases, when lockdown goes up by 1 standard deviation, daily covid-19 cases go down by 0.776 standard deviations (95% CI = −1.060 to −0.486; *p* < 0.05). From Table 7 and Figure 4, we can see that only lockdown and procedure have both direct and indirect effects. All other variables only have either a direct or indirect effect.

### 3.3. Path Diagram Layers

The path diagram presented in Figure 4 can be divided into five layers. Each layer consists of exogenous and endogenous variables. The five layers are constructed as follows:Layer 1 (L1) consists of family number, event, density, experience as exogenous variables, and medical resources as an endogenous variable.Layer 2 (L2) consists of family number, procedure, density, and experience as exogenous variables and economic resources as an endogenous variable.Layer 3 (L3) consists of event, procedure, density, experience as exogenous variables and lockdown as endogenous variable.Layer 4 (L4) consists of lockdown and economic resources as exogenous variables and technology used as an endogenous variable.Layer 5 (L5) consists of lockdown and technology used as exogenous variables and daily COVID-19 cases as an endogenous variable.

We divided the path diagram into five layers to better understand the indirect effects of all factors on daily COVID-19 infections.

For all endogenous variables, total effects are calculated as the sum of direct and indirect effects:Total effect = direct effect + indirect effect

All results obtained in Table 8 are calculated by the sum of results from Table 9 and Table 10.

The medical resources, lockdown, economic resources, and technology used are considered as both endogenous and intermediate variables.

Table 8, Table 9 and Table 10 represent the total, direct, and indirect effects of exogenous variables on intermediate variables, respectively.

## 4. Discussion

Daily COVID-19 infections are associated with the social and economic situation in each country and with level of each individual’s participation in society. Commitment to mitigation measures may have an impact as well. The framework of predicting daily COVID-19 cases is wide. A previous study showed that the number of diagnostic tests conducted positively affect the confirmed daily cases of COVID-19 [38]. Moreover, a path analysis was done on geographical determinants of COVID-19 daily infections in the U.S. [39]. Some studies tried to predict daily infections (dependent) using multiple linear regression on positive, deceased, and recovered cases (independent) [40]. Another study conducted in Italy showed that the mobility of citizens affected the recorded daily cases using multiple linear regression models [41]. Until now, no path analysis has been conducted to detect or assess the determinants of COVID-19 daily infections. Path analysis was used in this study to test a hypothesized model of daily COVID-19 cases in four different countries with different times to guide practice and provide directions for future research. Path analysis is superior to ordinary regression analysis as it provides an explanation of both the casual relation and the relative importance of alterative paths of influence [30]. We found that only lockdown and procedure have both direct and indirect effects on the rate of daily COVID-19 infections.

### 4.1. Direct Effects

The path coefficients showed that lockdown, technology used, and procedure have direct effects on COVID-19 daily infections. The largest impact is for the procedure and lockdown variables. An increase of 1 standard deviation in procedure degree leads to a decrease of 0.823 standard deviations in the COVID-19 daily infections and an increase of 1 standard deviation in lockdown degree produce a decrease of 0.776 standard deviations in COVID-19 daily infections. This highlights the importance of the commitment of every individual to the mitigation measures set in place by the authorities. The results also support the wearing of masks and social distancing, which help reduce the spread of COVID-19, thus reducing the daily confirmed cases. In terms of technology, COVID-19 daily infections increase by 0.17 standard deviations with the increase of 1 standard deviation in the used technology. This model shows that mitigation measures directly reduce the spread of COVID-19. The lack of direct effects from medical resources, experience, economic resources to COVID-19 daily cases are an unexpected, unique finding in this study. Future models could include other factors to assess and incorporate with the current model such as citizen movements in terms of foreign flights and local transportation.

### 4.2. Indirect Effects

From the results obtained in Table 7, Table 8, Table 9 and Table 10, all exogenous variables except technology used have indirect effects on daily COVID-19 infections from different patterns and routes. Moreover, all factors affect the daily cases indirectly through all intermediate variables except procedure. Procedure has an indirect effect on daily cases through lockdown, economic resources, and technology used, whereas all remaining exogenous variables include medical resources in their intermediate variables list. 

The final model shows that experience, special events, family number, and density have a significantly negative indirect effect on COVID-19 daily cases through their effects on lockdown. Lockdowns appear to be most strongly affected by procedure among the other exogenous variables. Medical and economic resources have significantly positive indirect effects on COVID-19 daily cases through their effects on technology used and lockdown degree. Technology used appears to be most strongly affected by economic resources and experience.

## 5. Conclusions

This study aimed to identify and assess the different socio-economic and mitigation measure determinants on COVID-19 daily cases. The study helped us detect some important factors to build an international effective strategy in the war against the COVID-19 pandemic. The findings, through the above analysis, indicate that implementing full lockdowns and the commitment to wearing masks and social distancing are essential for reducing daily COVID-19 infection rates. All other factors used in the study still have significant effects with different strengths and proportions.

There are still some limitations to this study. The data used comes from the beginning of the pandemic, which may not reflect today’s reality. Moreover, the data are a combination of several countries grouped together, which in turn differ in terms of area, population, economic, technological, and cultural capabilities. The limitations mentioned above reduce the generalizability of the findings in this study. 

Future studies can conduct path analysis on the determinants of the death rate caused by COVID-19 to help limit deaths and save more lives.

## Figures and Tables

**Figure 1 ijerph-18-10071-f001:**
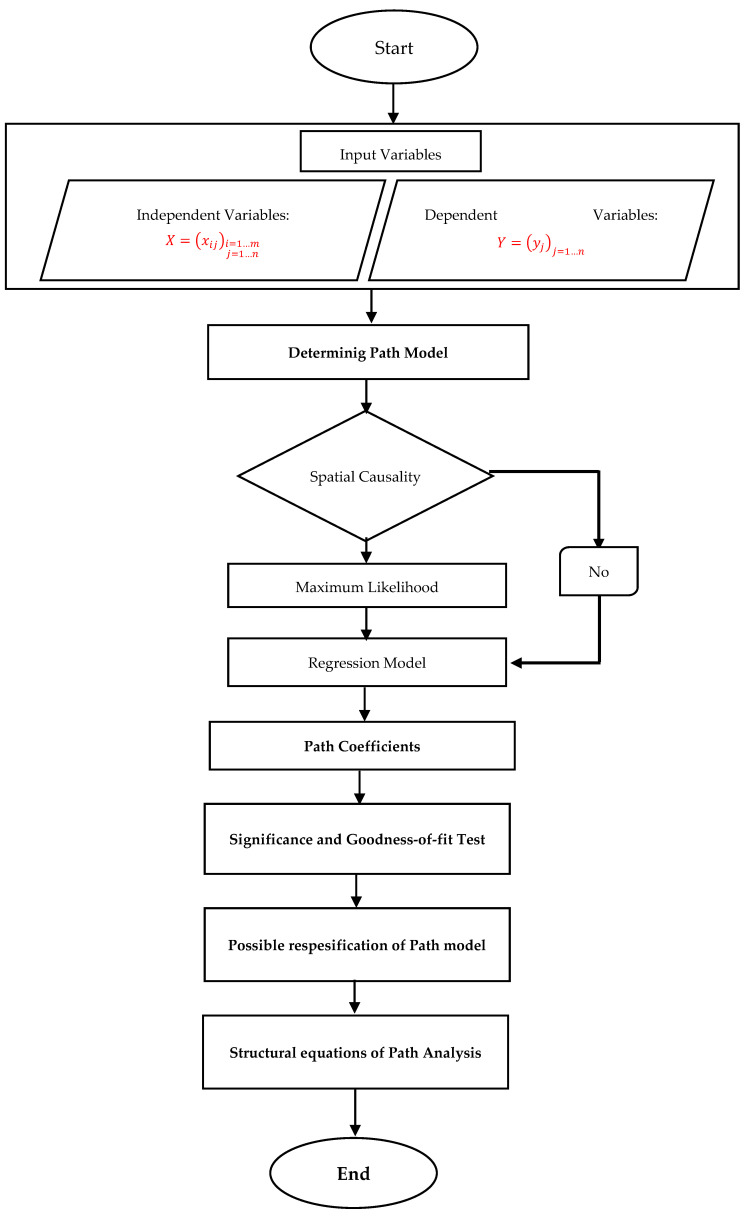
The stages of spatial path analysis.

**Figure 2 ijerph-18-10071-f002:**
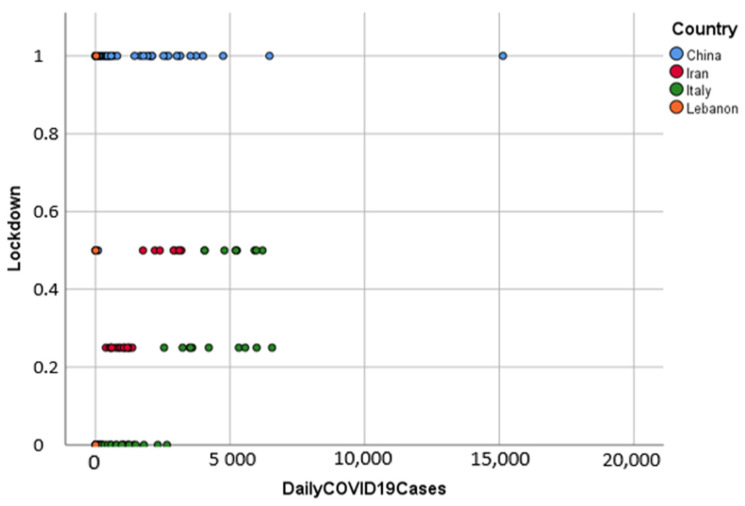
Grouped scatterplot of lockdown and daily Covid-19 cases by country.

**Figure 3 ijerph-18-10071-f003:**
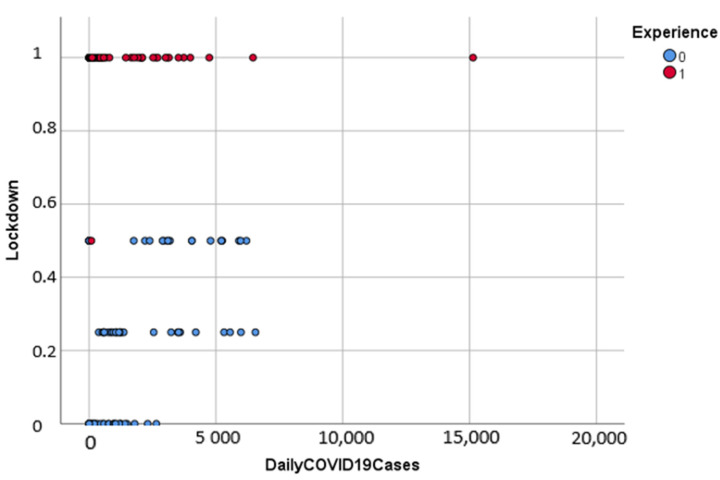
Grouped scatterplot of lockdown and daily Covid-19 cases by experience.

**Figure 4 ijerph-18-10071-f004:**
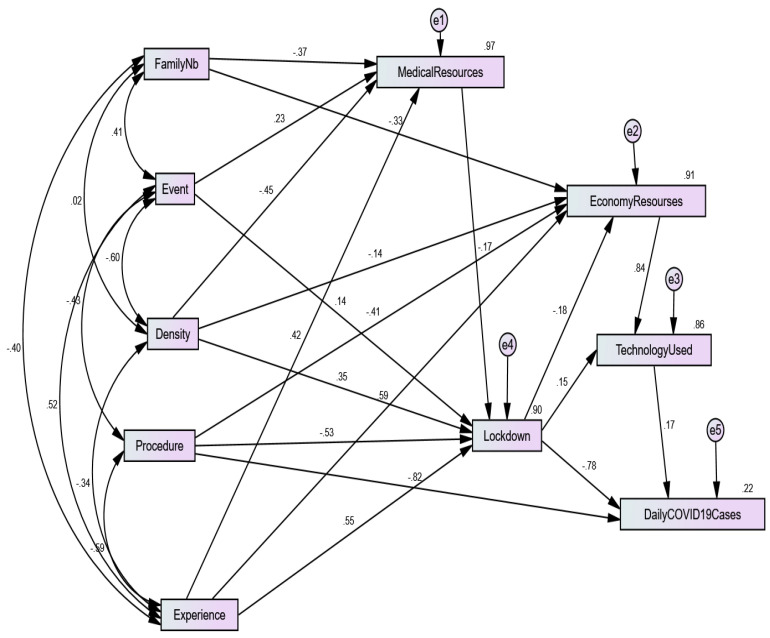
Path diagram of the default model with standardized parameter estimates.

**Table 1 ijerph-18-10071-t001:** The direct and indirect factors that are hypothesized to affect the spread level of COVID-19 in the selected countries.

Effective Factors	Indications	Values
Governance reactions (Lockdown)	No lockdown measures at all (value = 0).	0, 0.25, 0.5, 0.75, and 1
	Disabling major facilities: Stop flights at airports, public transport, ports, tourist places, universities and school closures and restrictions in movement in the public places (value = 0.25)	
	Partially lockdown measures (value= 0.5)	
	Fully lockdown measures (value= 0.75)	
	Providing virus checks abundantly (value = 1)	
Medical resources	0 = no medical resources, 0.25 = low, 0.5 = medium, 0.75 = good, 1 = high	0, 0.25, 0.5, 0.75, and 1
	Number of nurses for 1000 people	
	Number of doctors for 1000 people	
	Number of ICU beds for 1000	
Previous experience history	Yes = 1/no = 0	1/0
The used technology	0 = no technology, 0.25 = low, 0.5 = medium, 0.75 = good, 1 = high	0, 0.25, 0.5, 0.75, and 1
Special events	Yes = 1/no = 0	1/0
Economic resources and governmental aids	0 = no economic resources, 0.25 = low, 0.5 = medium, 0.75 = good, 1 = high	0, 0.25, 0.5, 0.75, and 1
Population density/km^2^	*	Number
Family number	*	Number
Quarantine commitment (Procedure)	1 = no commitment, 0.75 = low, 0.5 = medium, 0.25 = good, 0 = high	0, 0.25, 0.5, 0.75, and 1
The infected patients by the COVID-19	- new confirmed daily cases	Number

**Table 2 ijerph-18-10071-t002:** Summary statistics for the variables used in path analysis.

Descriptive Statistics					
A	N	Minimum	Maximum	Mean	Std. Deviation
Technology Used	210	0	1.00	0.4679	0.38339
Special Events	210	0	1	0.51	0.501
Procedure	210	0	1.00	0.3881	0.36924
Experience	210	0	1	0.32	0.467
Density	210	52	667	243.55	212.641
Family Number	210	3	5	3.59	0.803
Lockdown	210	0	1	0.5298	0.44292
Medical Resources	210	0.5	1	0.7774	0.18011
Economic resources	210	0.25	1	0.5405	0.33286
Daily COVID-19 Cases	210	0	15,136	1106.86	1866.425

**Table 3 ijerph-18-10071-t003:** Pearson correlation between the factors and the dependent variable.

	Technology Used	Event	Procedure	Experience	Density	Family Number	Temperature	Lockdown	Medical Resources	Economy Resources	Daily COVID-19 Cases
Technology Used	1	0.370 **	−0.710 **	0.845 **	−0.287 **	−0.540 **	0.410 **	0.671 **	0.762 **	0.931 **	0.235 **
		0	0	0	0	0	0	0	0	0	0.001
Event	0.370 **	1	−0.338 **	0.494 **	−0.620 **	0.334 **	0.161 *	0.282 **	0.574 **	0.339 **	0.088
	0		0	0	0	0	0.019	0	0	0	0.204
Procedure	−0.710 **	−0.338 **	1	−0.645 **	−0.012	0.224 **	−0.546 **	−0.865 **	−0.439 **	−0.676 **	−0.270 **
	0	0		0	0.86	0.001	0	0	0	0	0
Experience	0.845 **	0.494 **	−0.645 **	1	−0.318 **	−0.505 **	0.318 **	0.717 **	0.848 **	0.901 **	0.041
	0	0	0		0	0	0	0	0	0	0.558
Density	−0.287 **	−0.620 **	−0.012	−0.318 **	1	0.024	0.286 **	0.208 **	−0.711 **	−0.356 **	−0.245 **
	0	0	0.86	0		0.732	0	0.002	0	0	0
Family Number	−0.540 **	0.334 **	0.224 **	−0.505 **	0.024	1	0.048	−0.265 **	−0.509 **	−0.645 **	−0.152 *
	0	0	0.001	0	0.732		0.485	0	0	0	0.028
Temperature	0.410 **	0.161 *	−0.546 **	0.318 **	0.286 **	0.048	1	0.574 **	0.041	0.299 **	−0.032
	0	0.019	0	0	0	0.485		0	0.554	0	0.648
Lockdown	0.671 **	0.282 **	−0.865 **	0.717 **	0.208 **	−0.265 **	0.574 **	1	0.372 **	0.629 **	0.045
	0	0	0	0	0.002	0	0		0	0	0.512
Medical Resources	0.762 **	0.574 **	−0.439 **	0.848 **	−0.711 **	−0.509 **	0.041	0.372 **	1	0.839 **	0.198 **
	0	0	0	0	0	0	0.554	0		0	0.004
Economic Resources	0.931 **	0.339 **	−0.676 **	0.901 **	−0.356 **	−0.645 **	0.299 **	0.629 **	0.839 **	1	0.253 **
	0	0	0	0	0	0	0	0	0		0
Daily COVID-19 Cases	0.235 **	0.088	−0.270 **	0.041	−0.245 **	−0.152 *	−0.032	0.045	0.198 **	0.253 **	1
	0.001	0.204	0	0.558	0	0.028	0.648	0.512	0.004	0	

** Correlation is significant at the 0.01 level (two-tailed). * Correlation is significant at the 0.05 level (two-tailed).

**Table 4 ijerph-18-10071-t004:** Correlation estimates between exogenous variables.

Correlation			Estimate
Procedure	<-->	Experience	−0.593
Density	<-->	Experience	−0.340
Family Number	<-->	Experience	−0.396
Family Number	<-->	Event	0.415
Event	<-->	Density	−0.597
Event	<-->	Procedure	−0.426
Event	<-->	Experience	0.521
Family Number	<-->	Density	0.024

**Table 5 ijerph-18-10071-t005:** Squared multiple correlations.

Dependent Variable	Estimate (R^2^)
Medical Resources	0.972
Lockdown	0.903
Economic Resources	0.908
Technology Used	0.864

**Table 6 ijerph-18-10071-t006:** Estimated parameters for all factors. *** *p*-Value < 0.001.

Dependent Variable		Independent Variable	Standardized Parameter Estimate	*p*-Value
Medical Resources	<---	Family Number	−0.366	***
Medical Resources	<---	Event	0.229	***
Medical Resources	<---	Density	−0.449	***
Medical Resources	<---	Experience	0.418	***
Lockdown	<---	Medical Resources	−0.175	0.052
Lockdown	<---	Event	0.145	***
Lockdown	<---	Density	0.346	***
Lockdown	<---	Procedure	−0.532	***
Lockdown	<---	Experience	0.552	***
Economic Resources	<---	Family Number	−0.331	***
Economic Resources	<---	Density	−0.143	***
Economic Resources	<---	Procedure	−0.410	***
Economic Resources	<---	Experience	0.589	***
Economic Resources	<---	Lockdown	−0.181	0.007
Technology Used	<---	Lockdown	0.149	***
Technology Used	<---	Economic Resources	0.837	***
Daily COVID-19 Cases	<---	Technology Used	0.170	0.035
Daily COVID-19 Cases	<---	Lockdown	−0.775	***
Daily COVID-19 Cases	<---	Procedure	−0.823	***

**Table 7 ijerph-18-10071-t007:** Path Analysis on socio-economic and mitigation measure determinants of COVID-19 daily infections.

Dependent Variable		Independent Variable	Standardized Effect Estimate	L.B	U.B	*p*-Value
Indirect Effect				95% CI		
Daily COVID-19 Cases	<---	Procedure	0.355	0.190	0.602	0.003
Daily COVID-19 Cases	<---	Lockdown	−0.001	−0.031	0.050	0.976
Daily COVID-19 Cases	<---	Experience	−0.288	−0.475	−0.122	0.003
Daily COVID-19 Cases	<---	Family Number	−0.097	−0.155	−0.023	0.022
Daily COVID-19 Cases	<---	Density	−0.35	−0.493	−0.231	0.003
Daily COVID-19 Cases	<---	Event	−0.081	−0.215	0.022	0.120
Daily COVID-19 Cases	<---	Medical Resources	0.136	−0.050	0.274	0.110
Daily COVID-19 Cases	<---	Economic Resources	0.142	0.011	0.236	0.034
**Direct Effect**						
Daily COVID-19 Cases	<---	Technology Used	0.17	0.015	0.287	0.031
Daily COVID-19 Cases	<---	Lockdown	−0.775	−10.051	−0.497	0.006
Daily COVID-19 Cases	<---	Procedure	−0.823	−10.175	−0.541	0.004
**Total Effect = Direct Effect + Indirect Effect**						
Daily COVID-19 Cases	<---	Lockdown	−0.776	−10.060	−0.486	0.005
Daily COVID-19 Cases	<---	Procedure	−0.468	−0.649	−0.314	0.003

**Table 8 ijerph-18-10071-t008:** Standardized total effects of exogenous variables on endogenous variables.

	Experience	Procedure	Density	Event	Family Number	Medical Resources	Lockdown	Economic Resources	Technology Used
Medical Resources	0.418	0.000	−0.449	0.229	−0.366	0.000	0.000	0.000	0.000
Lockdown	0.479	−0.532	0.425	0.105	0.064	−0.175	0.000	0.000	0.000
Economic Resources	0.503	−0.314	−0.220	−0.019	−0.342	0.032	−0.181	0.000	0.000
Technology Used	0.492	−0.342	−0.121	0.000	−0.277	0.000	−0.003	0.837	0.000

**Table 9 ijerph-18-10071-t009:** Standardized direct effects of exogenous variables on endogenous variables.

	Experience	Procedure	Density	Event	Family Number	Medical Resources	Lockdown	Economic Resources	Technology Used
Medical Resources	0.418	0.000	−0.449	0.229	−0.366	0.000	0.000	0.000	0.000
Lockdown	0.552	−0.532	0.346	0.145	0.000	−0.175	0.000	0.000	0.000
Economic Resources	0.589	−0.410	−0.143	0.000	−0.331	0.000	−0.181	0.000	0.000
Technology Used	0.000	0.000	0.000	0.000	0.000	0.000	0.149	0.837	0.000

**Table 10 ijerph-18-10071-t010:** Standardized indirect effects of exogenous variables on endogenous variables.

	Experience	Procedure	Density	Event	Family Number	Medical Resources	Lockdown	Economic Resources	Technology Used
Medical Resources	0.000	0.000	0.000	0.000	0.000	0.000	0.000	0.000	0.000
Lockdown	−0.073	0.000	0.079	−0.040	0.064	0.000	0.000	0.000	0.000
Economic Resources	−0.087	0.096	−0.077	−0.019	−0.012	0.032	0.000	0.000	0.000
Technology Used	0.492	−0.342	−0.121	0.000	−0.277	0.000	−0.151	0.000	0.000

## Data Availability

All data, models, and code generated or used during the study appear in the submitted article.

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
