# Peer review of "Path Analysis to Assess Socio-Economic and Mitigation Measure Determinants for Daily Coronavirus Infections"

_ijerph, 2021, doi:10.3390/ijerph181910071_

Round 1

Reviewer 1 Report

This is a novel publication. A lot has been and is been written on COVID-19 but this work provides a clear co-relation between the socio-economic organization of select countries against mitigation measures.

Author Response

In reply to the submission of the manuscript 

“Path Analysis to assess Socio-economic and Mitigation Measure Determinants for Daily Coronavirus Infections”

Dear Reviewer
First, we would like to thank the reviewers for their constructive remarks that improve the quality of this article. 
We are pleased to inform you that the proposed article has been modified considering the point of view of the reviewers. In the following, questions from reviewers are in bold font and our comments are provided in normal black font. The corrections that we propose have been included in the article in normal red font.

Best regards,
The authors, 
Abbas Rammal, Elie Yammine.

Reviewer 2 Report

Thank you for the opportunity to review this study.

I think that timing of lockdown and severity of lockdown as factors predicting number of cases (but more importantly number of deaths) is of great importance.

A very robust statistical process (which could be more clearly articulated) has been followed, but this study is lacking in the theoretical development to inform this analysis.

It is recommended that this study could be greatly improved with a clear conceptual framework grounded in the academic literature to develop a model which distinguishes between the independent variables of interest and confounding variables that need to be held constant and/or other dependencies – such as the effect of economic variables on timing and strictness of lockdown.

Ultimately, given the theoretical limitations of this study and the lack of clarity in its methods I do not feel that it is of a strong enough standard for publication in this journal.

My specific comments are:

• Abstract refers to relatively high daily cases recorded and such high cases – but relative to what? And what is your criteria for high?

• How will identifying the key factors that contribute to coronavirus infections remove this “danger and continuous threat”? For example, if we know that population density and whether or not a state is an island are strongly predictive of infections, how will this knowledge suddenly remove danger and continuous threat?

• The trends referred to are very dated – up to June 2020 – please update and use more comparable data such as death rate, rather than total cases, which are subject to population and testing capacity.

• Timing of introduction of social distancing and severity of social distancing are presented as factors that have varied, suggesting these are your independent variables for the study.

• What is meant by “benefit of global expectation”?

• “behavior of Coronavirus infections” and “mutation of daily infections” – consider more appropriate terminology.

• The stated aim of “examining socio-economic and mitigation measure determinants of daily infections” needs to be fleshed out – you have introduced timing of introduction of social distancing and severity of social distancing as important variables, but what is meant by socio-economic determinants and how would these be modelled – for example, are socio-economic factors predictors of a nations timing and severity of lockdown? For example, Colombia implemented a lockdown at the start of the pandemic but even in the face of cases being at their peak as of July 2021, there is not the economic resources (or potentially political will) to enforce any kind of lockdown.

• Greater clarity is needed about the relationship between variables before the authors can conclude that path analysis is the appropriate methodology by which study the hypothesised relationships.

• The description of path analysis as a form of multiple regression, without referring to directed dependancies between variables, does not provide the information needed to support your conclusion that path analysis is appropriate to meet the aim of your study.

• Indeed, I would much prefer to clearly understand your conceptual framework and then you can justify your choice of path analysis (i.e., a series of multiple regressions so to establish directed dependencies between variables) in your methodology section.

• Related to this, your review of literature on studies using path analysis is not important – you should focus on your variables and literature on these variables to establish the conceptual framework you are going to test. • Your failure to do this is a major flaw in this study.

Materials and Methods - Data set collection

• “The essential pavement procedure to answer and judge the test hypothesis and evaluating the outcomes of some particular collections is the collecting and measuring process of data.” What does this mean?

• You need to provide greater justification for your choice of countries – you state that these countries adopted different measures – but can you operationalize these as variables to test the effect of timing of introduction of social distancing and severity of social distancing?

• What do the different number of days in brackets after each country represent?

• You introduce several indicators in your methodology, but these needed to be covered in your introduction and articulated as a conceptual framework.

• You refer to these factors as direct and indirect – but again, this needs to be articulated as a conceptual framework.

• Rather than thinking about direct and indirect factors, I think you should clearly define independent variables of interest (e.g., timing of and severity of lockdowns) and then potential confounding variables that you are holding constant in your modelling.

• Much more detail is needed about how the values of 0, 0.25, 0.5. 0.75 and 1 are attributed to each variable.

• The dependent variable is daily infections, but is not clear how you are operationalising this.

• Is it that you have recorded daily infections over 210 days for each country, so that you have a sample size of 840 days? This is not clear. Are you controlling for country data was collected from?

• “These direct and indirect factors allow to measure the score of each sample which 143 represents the times in days in four countries: China, Iran, Italy, and Lebanon. These sam- 144 ples were classified by 3 categories which are the degrees of possibility of Coronavirus 145 disease infection according to their evolution (in days): Low, medium, and High. Table 1 146 shows the coding of each of the factors used to build the model.” What does all of this mean?

• You refer to path analysis, SEM and spatial path analysis – greater clarity in the relationships between these approaches is required.

• Results - The distribution in days is confusing – was 210 days of data not collected from each country?

Author Response

Response to the reviewers’ comments

In reply to the submission of the manuscript

“Path Analysis to assess Socio-economic and Mitigation Measure Determinants for Daily Coronavirus Infections”

Dear Reviewers and associate editor.

First, we would like to thank the reviewers for their constructive remarks that improve the quality of this article.

We are pleased to inform you that the proposed article has been modified considering the point of view of the reviewers. In the following, questions from reviewers are in bold font and our comments are provided in normal black font. The corrections that we propose have been included in the article in normal red font.

Best regards,

The authors,

Abbas Rammal, Elie Yammine.

reviewer's comments:

Comment #1: Abstract refers to relatively high daily cases recorded and such high cases – but relative to what? And what is your criteria for high?

Answer: Your remark is very important. We added the following explanation “recorded in a short time compared to other seasonal flu” and “high” is changed to “unexpected”.

Comment #2: How will identifying the key factors that contribute to coronavirus infections remove this “danger and continuous threat”? For example, if we know that population density and whether or not a state is an island are strongly predictive of infections, how will this knowledge suddenly remove danger and continuous threat?

Answer: We have added the following explanation in the abstract “and can guide national authorities to strict or reduce the mitigation measures, in addition to spreading awareness and attention to the important factors which contribute to the propagation of the disease”

Comment #3: The trends referred to are very dated – up to June 2020 – please update and use more comparable data such as death rate, rather than total cases, which are subject to population and testing capacity.

Answer: Your comment is very relevant. Despite the importance of knowing the determinants of death rates or any other feature, we are trying in this study to know what are the socio-economic factors that mostly contribute to the spread of covid-19 and that is why our dependent variable is the total infections of the disease. For future work, we added a new horizon in the Conclusion Section “Further studies can conduct path analysis on the determinants of the death rate caused by Covid-19 to help limit the deaths and save more lives”

Comment #4: Timing of introduction of social distancing and severity of social distancing are presented as factors that have varied, suggesting these are your independent variables for the study.

Answer: In this study we did not include the timing of introduction of social distancing and severity of social distancing as factors, instead we include degree of intervention of local authorities in each country ad the social distancing is used as mitigation measure for countries to reduce the daily cases of covid-19 disease.

Comment #5: What is meant by “benefit of global expectation”?

Answer: This mistake is corrected “benefit of global health security”

Comment #6: “behavior of Coronavirus infections” and “mutation of daily infections” – consider more appropriate terminology.

Answer: This comment has been made.

Comment #7: The stated aim of “examining socio-economic and mitigation measure determinants of daily infections” needs to be fleshed out – you have introduced timing of introduction of social distancing and severity of social distancing as important variables, but what is meant by socio-economic determinants and how would these be modelled – for example, are socio-economic factors predictors of a nations timing and severity of lockdown? For example, Colombia implemented a lockdown at the start of the pandemic but even in the face of cases being at their peak as of July 2021, there is not the economic resources (or potentially political will) to enforce any kind of lockdown.

Answer: As mentioned in comment #4 we did not use social distancing as a predictor. We intended to determine the important factors that contribute to the most in the daily cases of covid-19 disease. The socio-economic factors used are clearly stated in the Data Description section with how each variable takes its values and there is no social distance factor among them. Lockdown is among the predictors taking values between 0 and 1 as 0 with no lockdown implemented and 1 as a full-lockdown.

Comment #8: Greater clarity is needed about the relationship between variables before the authors can conclude that path analysis is the appropriate methodology by which study the hypothesized relationships.

Answer: Your remark is very important. For that, we added in the Results section a table of Pearson correlations between independent variables and the dependent variable (covid-19 daily cases).

TechnologyUsed

Event

Procedure

Experience

Density

FamilyNb

Temperature

Lockdown

MedicalResources

EconomyResourses

DailyCOVID19Cases

TechnologyUsed

1

.370**

-.710**

.845**

-.287**

-.540**

.410**

.671**

.762**

.931**

.235**

0

0

0

0

0

0

0

0

0

0.001

Event

.370**

1

-.338**

.494**

-.620**

.334**

.161*

.282**

.574**

.339**

0.088

0

0

0

0

0

0.019

0

0

0

0.204

Procedure

-.710**

-.338**

1

-.645**

-0.012

.224**

-.546**

-.865**

-.439**

-.676**

-.270**

0

0

0

0.86

0.001

0

0

0

0

0

Experience

.845**

.494**

-.645**

1

-.318**

-.505**

.318**

.717**

.848**

.901**

0.041

0

0

0

0

0

0

0

0

0

0.558

Density

-.287**

-.620**

-0.012

-.318**

1

0.024

.286**

.208**

-.711**

-.356**

-.245**

0

0

0.86

0

0.732

0

0.002

0

0

0

FamilyNb

-.540**

.334**

.224**

-.505**

0.024

1

0.048

-.265**

-.509**

-.645**

-.152*

0

0

0.001

0

0.732

0.485

0

0

0

0.028

Temperature

.410**

.161*

-.546**

.318**

.286**

0.048

1

.574**

0.041

.299**

-0.032

0

0.019

0

0

0

0.485

0

0.554

0

0.648

Lockdown

.671**

.282**

-.865**

.717**

.208**

-.265**

.574**

1

.372**

.629**

0.045

0

0

0

0

0.002

0

0

0

0

0.512

MedicalResources

.762**

.574**

-.439**

.848**

-.711**

-.509**

0.041

.372**

1

.839**

.198**

0

0

0

0

0

0

0.554

0

0

0.004

EconomyResourses

.931**

.339**

-.676**

.901**

-.356**

-.645**

.299**

.629**

.839**

1

.253**

0

0

0

0

0

0

0

0

0

0

DailyCOVID19Cases

.235**

0.088

-.270**

0.041

-.245**

-.152*

-0.032

0.045

.198**

.253**

1

0.001

0.204

0

0.558

0

0.028

0.648

0.512

0.004

0

**. Correlation is significant at the 0.01 level (2-tailed)

*. Correlation is significant at the 0.05 level (2-tailed)

We also added an explanation “Table 2 shows that there exist 5 variables (TechnologyUsed, Procedure, Density, MedicalResources, EconomicResources) that are correlated with the daily cases at a 0.01 level of significance, while only one variable (Family NB) is correlated to the daily cases with 0.05 level of significance. Having 6 out of 10 variables that are significantly correlated to the daily cases of covid-19, and since we are trying to assess and detect the factors that most contribute to the daily infections, thus path analysis is the right methodology to use.”

Comment #9: The description of path analysis as a form of multiple regression, without referring to directed dependancies between variables, does not provide the information needed to support your conclusion that path analysis is appropriate to meet the aim of your study.

Answer: Your comment is relevant. For that, we added an explanation in the Results section “Having 6 out of 10 variables that are significantly correlated to the daily cases of covid-19, and since we are trying to assess and detect the factors that most contribute to the daily infections, thus path analysis is the right methodology to use. Most variables are correlated to each other which creates dependencies and associations among the independent variables, thus mediators (variables carrying the indirect effects) have high correlations with the independent ones.”

Comment #10: Indeed, I would much prefer to clearly understand your conceptual framework and then you can justify your choice of path analysis (i.e., a series of multiple regressions so to establish directed dependencies between variables) in your methodology section.

Answer: Your comment is very important. We have added a conceptual framework sub-section in the materials and methods section explaining the importance of the variables used and the connections between each other.

Comment #11: Related to this, your review of literature on studies using path analysis is not important – you should focus on your variables and literature on these variables to establish the conceptual framework you are going to test. • Your failure to do this is a major flaw in this study.

Answer: As mentioned in comment #10, the problem is resolved.

Comment #12: “The essential pavement procedure to answer and judge the test hypothesis and evaluating the outcomes of some particular collections is the collecting and measuring process of data.” What does this mean?

Answer: This phrase is corrected to “The mechanism to answer and judge the test hypothesis and evaluate the outcomes of particular questions is the collecting and measuring process of data”

Comment #13: You need to provide greater justification for your choice of countries – you state that these countries adopted different measures – but can you operationalize these as variables to test the effect of timing of introduction of social distancing and severity of social distancing?

Answer: We did not include the country as an independent variable, instead we collected the data from these countries relying that these countries have different social values and economies. We tried to vary our selection of countries from different continents and cultures to have a variety in the data. Collecting data from neighbor countries will not be useful for our study. We repeat that we did not include social distancing as an independent variable but it was used in the context as a mitigation measure. We added an extra clarification “We tried to vary as much as we can the selection of the countries based on social values and economies and different cultures”.

Comment #14: What do the different number of days in brackets after each country represent?

Answer: Those numbers are the number of days collected in every country with an aggregate of 210 days in total. For example, 67 days are displayed for China whereas 40 days are only collected from Lebanon. Overlaps may occur in dates i.e., specific days collected in China may also been collected in Lebanon with some additional days.

Comment #15: You introduce several indicators in your methodology, but these needed to be covered in your introduction and articulated as a conceptual framework.

Answer: This comment has been resolved. A Conceptual Framework section has been inserted in the Materials and Methods section. Also, a summary on the variables used is inserted in the Introduction section.

Comment #16: You refer to these factors as direct and indirect – but again, this needs to be articulated as a conceptual framework.

Answer: This comment has been resolved.

Comment #17: Rather than thinking about direct and indirect factors, I think you should clearly define independent variables of interest (e.g., timing of and severity of lockdowns) and then potential confounding variables that you are holding constant in your modelling.

Answer: Your comment is relevant. From a statistical point of view, we did not apply a simple regression model in our study so that we only have independent variables and a dependent variable. Instead, we used path analysis which beyond regression analysis it can determine what are the mediator variables connecting the dependent variable with the independent variables. Path analysis do not include fixing some variables at a given constant, it rather finds what are the mediators of the model. Path analysis is more complicated than regression analysis but it gives better results. So, using path analysis we will have direct and indirect effects from the independent variables to the dependent variable where the indirect effects pass through the mediators.

Comment #18: Much more detail is needed about how the values of 0, 0.25, 0.5. 0.75 and 1 are attributed to each variable.

Answer: Your remark is important. We already explained how values are attributed to the variables in Table 1. With 0 representing the absence of the variable and the values 0.25 = low presence, 0.5 =medium presence, 0.75 = good presence, till 1 which represents the full presence of the variable.

Comment #19: The dependent variable is daily infections, but is not clear how you are operationalising this.

Answer: Your comment is very relevant. The dependent variable is daily infections records which are basically the cumulative records for a dependent day-by-day scale. In other words, the record of the next day is the sum of the records of the current day and the new records obtained in the same day.

Comment #20: Is it that you have recorded daily infections over 210 days for each country, so that you have a sample size of 840 days? This is not clear. Are you controlling for country data was collected from?

Answer:

The date range for the data for each of the four countries is as the following:

  • China: 9 January 2020 – 28 March 2020 (80 days)
  • Lebanon: 21 February 2020 – 31 March 2020 (40 days)
  • Italy: 31 January 2020 – 31 March 2020 (61 days)
  • Iran: 19 February 2020 – 31 March 2020 (42 days)

Comment #21: “These direct and indirect factors allow to measure the score of each sample which represents the times in days in four countries: China, Iran, Italy, and Lebanon. These samples were classified by 3 categories which are the degrees of possibility of Coronavirus disease infection according to their evolution (in days): Low, medium, and High. Table 1 shows the coding of each of the factors used to build the model.” What does all of this mean?

Answer: Your comment is very important, for this we have deleted the following sentence: "These direct and indirect factors allow to measure the score of each sample which represents the times in days in four countries: China, Iran, Italy, and Lebanon. were classified by 3 categories which are the degrees of possibility of Coronavirus disease infection according to their evolution (in days): Low, medium, and High. "

We explained the mean of "Table 1 shows the coding of each of the factors used to build the model” by: “The Table 1 shows the technique for coding each of the factors by developing the measurement scales used to build the model. That's the first basic step procedure to build and develop a model by the Structural Equation Modelling method.”

Comment #22: You refer to path analysis, SEM and spatial path analysis – greater clarity in the relationships between these approaches is required.

Answer: Your comment is very relevant. The SEM is a powerful, multivariate technique found increasingly in scientific investigations to test and evaluate multivariate causal relationships. SEMs differ from other modeling approaches as they test the direct and indirect effects on pre-assumed causal relationships. Path analysis was developed to quantify the relationships among multiple variables. It was the early name for SEM before there were latent variables and was very powerful in testing and developing the structural hypothesis with both indirect and direct causal effects. However, the two effects have recently been synonymized. Path analysis can explain the causal relationships among variables. A common function of path analysis is mediation, which assumes that a variable can influence an outcome directly and indirectly through another variable.

Comment #23: Results - The distribution in days is confusing – was 210 days of data not collected from each country?

Answer:

The distribution of 210 days of data was in the following form:

  • China: 9 January 2020 – 28 March 2020 (80 days)
  • Lebanon: 21 February 2020 – 31 March 2020 (40 days)
  • Italy: 31 January 2020 – 31 March 2020 (61 days)
  • Iran: 19 February 2020 – 31 March 2020 (42 days)

Reviewer 3 Report

Although the study is specifically related with Covid-19 mitigation measures adopted by the countries analysed, the path analysis could be replicated and applied to any future similar crisis. The use of tables and graphs will help students and non topic experts to read and navigate the paper.

A full bias analysis seem missing. It could be useful, in order to improve the scientific soundness of the paper, add to the paper the study's authors bias analysis.

Author Response

In reply to the submission of the manuscript

“Path Analysis to assess Socio-economic and Mitigation Measure Determinants for Daily Coronavirus Infections”

Dear Reviewers and associate editor.

First, we would like to thank the reviewers for their constructive remarks that improve the quality of this article.

We are pleased to inform you that the proposed article has been modified considering the point of view of the reviewers. In the following, questions from reviewers are in bold font and our comments are provided in normal black font. The corrections that we propose have been included in the article in normal red font.

Best regards,

The authors,

Abbas Rammal, Elie Yammine.

reviewer's comments:

Comment 1: Although the study is specifically related with Covid-19 mitigation measures adopted by the countries analysed, the path analysis could be replicated and applied to any future similar crisis. The use of tables and graphs will help students and non-topic experts to read and navigate the paper.

Answer: Your comment is relevant. In our paper, we used the tables and graphs to help students and non-topic experts to read and navigate the paper. You can show that in the paper. In addition, we added the following descriptive statistics table:

Descriptive Statistics

a

N

Minimum

Maximum

Mean

Std. Deviation

Technology Used

210

0

1.00

0.4679

0.38339

Special Event

210

0

1

0.51

0.501

Procedure

210

0

1.00

.3881

0.36924

Experience

210

0

1

0.32

0.467

Density

210

52

667

243.55

212.641

Family Number

210

3

5

3.59

0.803

Lockdown

210

0

1

0.5298

0.44292

Medical Resources

210

0.5

1

0.7774

0.18011

Economy Resources

210

0.25

1

0.5405

0.33286

DailyCOVID19Cases

210

0

15136

1106.86

1866.425

Also, these 2 additional graphs:

This graph shows that China which implemented high lockdown standards recorded covid-19 cases as much as other countries which implemented partial lockdowns. While Iran recorded less cases than Italy with the same lockdown measures (Lockdown = 0.25,0.5).

This graph shows that countries having experience with health crisis helps in reducing the transmission of covid-19 whereas countries with no previous experience recorded higher cases of the virus.

Comment 2: A full bias analysis seem missing. It could be useful, in order to improve the scientific soundness of the paper, add to the paper the study's authors bias analysis.

Answer: For the bias analysis, we have created a graphical display of presumed causal relations among variables and their measurements (Figure 2), and we have presented in the table 2 the Pearson correlation between the independent variables among each other and between every independent variable and the daily cases of covid-19. In addition, a systematic error or bias analysis is not applicable in our case since we are applying path analysis (regression and Structural equation modeling) on the covid-19 daily cases. No classification or any probabilistic model is used. Data collected are from different countries having variety of cultures and the days recorded are almost the same days in every country which minimizes the systematic error close to 0.
